# Comparison of C-Arm-Free Oblique Lumbar Interbody Fusion L5-S1 (OLIF51) with Transforaminal Lumbar Interbody Fusion L5-S1 (TLIF51) for Adult Spinal Deformity

**DOI:** 10.3390/medicina59050838

**Published:** 2023-04-26

**Authors:** Masato Tanaka, Sumeet Sonawane, Umesh Meena, Zhichao Lu, Yoshihiro Fujiwara, Takuya Taoka, Koji Uotani, Yoshiaki Oda, Tomoyoshi Sakaguchi, Shinya Arataki

**Affiliations:** 1Department of Orthopaedic Surgery, Okayama Rosai Hospital, Okayama 702-8055, Japan; drsumeet166@gmail.com (S.S.); gtbumesh@gmail.com (U.M.); luzhichao8753@hotmail.com (Z.L.); fujiwarayoshihiro2004@yahoo.co.jp (Y.F.); taoktakuya@gmail.com (T.T.); coji.uo@gmail.com (K.U.); odaaaaaaamn@yahoo.co.jp (Y.O.); araoyc@gmail.com (S.A.); 2Department of Orthopaedic Surgery, Okayama University Hospital, Okayama 700-8558, Japan; 3Department of Rehabilitation, Okayama Rosai Hospital, Okayama 702-8055, Japan; tomoyoshi0127@gmail.com

**Keywords:** adult spinal deformity, minimally invasive surgery, oblique lumbar interbody fusion, transforaminal lumbar interbody fusion, navigation, C arm free

## Abstract

*Background and Objectives*: Adult spinal deformity (ASD) surgery, L5-S1 lordosis is very important factor. The main objective of the research is to retrospectively compare symptomatic presentation and radiological presentation in the sequelae of oblique lumbar inter-body spinal fusion at L5-S1 (OLIF51) and transforaminal lumbar interbody fusion (TLIF) for ASD. *Materials and Methods*: We retrospectively evaluated 54 patients who underwent corrective spinal fusion for ASD between October 2019 and January 2021. Thirteen patients underwent OLIF51 (average 74.6 years old, group O) and 41 patients underwent TLIF51 (average 70.5 years old, group T). Mean follow-up period was 23.9 months for group O and 28.9 months for group T, ranging from 12 to 43 months. Clinical and radiographic outcomes are assessed using values including visual analogue scale (VAS) for back pain and Oswestry disability index (ODI). Radiographic evaluation was also collected preoperatively and at 6, 12, and 24 months postoperatively. *Results*: Surgical time in group O was less than that in group T (356 min vs. 492 min, *p* = 0.003). However, intraoperative blood loss of both groups were not significantly different (1016 mL vs. 1252 mL, *p* = 0.274). Changes in VAS and ODI were similar in both groups. L5-S1 angle gain and L5-S1 height gain in group O were significantly better than those of group T (9.4° vs. 1.6°, *p* = 0.0001, 4.2 mm vs. 0.8 mm, *p* = 0.0002)*. Conclusions*: Clinical outcomes were not significantly different in both groups, but surgical time in OLIF51 was significantly less than that in TLIF51. The radiographic outcomes showed that OLIF51 created more L5-S1 lordosis and L5-S1 disc height compared with TLIF 51.

## 1. Introduction

Adult spinal deformity (ASD) consists of spinal malalignment throughout adulthood [1,2]. Its previous is up to 68% in the elderly individuals aged >65 years [3]. ASD affect not only patients’ general health [4] but also psychological problems such as depression [5]. Conservative treatment for ASD has been regarded effective for slight malalignment, so surgical intervention is necessary for the severe cases [6]. However, deformity correction surgery for aged patients has enormous risk for them. Traditionally, open osteotomies like Ponte osteotomy, pedicle subtraction osteotomy (PSO), and vertebral column resection (VCR) have been performed to achieve good spinal alignment and excellent clinical outcomes [7,8]. However, these techniques have great number of complications such as pseudoarthrosis, focal sensory motor deficits, dural tear, deep surgical site infection, and excessive blood loss [9,10]. Until now, the high mortality rate of 2.4% and high complication rate up to 70% for ASD surgery have been reported [11,12].

After fusion surgery on the lumbar spine, it is essential to achieve strong fusion in order to have favorable postoperative results. Interbody operations, including those involving the insertion of a cage, are necessary for lumbar fusion surgery, and methods for interbody procedures have evolved significantly over the last few decades [13]. Conventionally, the gold-standard procedure for interbody fusion has been cage insertion via the transforaminal route after a posterior approach (transforaminal lumbar interbody spinal fusion, TLIF). Nevertheless, this surgical method has does present with known procedural complications such as injury to the nerve root, breach of the vertebral endplate, collapse of displacement of the inter body cage, and other perilous complications [13].

Therefore, minimally invasive surgery (MIS) for spinal fusion such as minimally invasive transforaminal lumbar interbody fusion (MIS-TLIF), oblique lumbar interbody fusion (OLIF), direct lateral lumbar interbody fusion (DLIF) and percutaneous pedicle screw fixation (PPS) have been developed and these techniques become world standard [14]. Recent studies have established that, intraoperative blood loss and preoperative complications occurring during ASD surgery have been markedly diminished by implementing the current MIS methods [15,16]. Anand et al., reported excellent results of circumferential minimally invasive surgery (c-MIS) for ASD [17]. With these MIS technique for ASD, the rate of occurrence of complications has been dramatically reduced [18]. In 2021, this c-MIS technique has been performed under navigation guidance without intraoperative fluoroscopy [14]. To reduce intraoperative exposure of the operating team to radiation, the use of which is otherwise inevitable for conventional surgical methods [19].

Another important key to achieve a successful ASD deformity corrective surgery is imperative to create adequate lumbar lordosis, especially in lower lumbar spine [2]. The angle between L4 and S1 creates more than two third of lumbar lordosis [20], so to get normal lumbar lordosis, it is very important to create adequate L5-S1 lordosis. MIS TLIF at L5-S1 is one of the options for this, but nowadays OLIF at L5-S1 area (OLIF51) is getting popular to create good L5-S1 lordosis [21].

The focus of this research has been to retrospectively compare clinical and radiographic outcomes of MIS-TLIF 51 and OLIF 51 techniques for ASD.

## 2. Materials and Methods

This research has been approved by the ethics committee of our institution (No. 396). The necessary informed consents were duly signed and obtained from all the patients involved in the study. A retrospective analysis of the cohort of 54 patients who underwent ASD corrective surgery at our institute during the time period October 2016 and January 2022.

The following Inclusion criteria are considered for this research (1) patients with an age of 60 years or more with the established occurrence of at least one of the conditions mentioned: the sagittal vertical axis (SVA) 95 mm or greater, pelvic tilt (PT) 30 degrees or more, with or without a coronal Cobb angle of 30 degrees or higher [3], (2) severe low back ache, difficulty of walking, and disturbance of active daily life, (3) failure to improve symptomatically following 2 months of conservative treatment. Exclusion criteria were deformities of the spine resulting from acute or chronic infections spine, or neoplasms of spine. The operative procedures were taken up in two stages, Primary OLIF L1 to L5 (or S1), Secondary corrective posterior spinal fusion spanning from T10 to pelvis (+MIS-TLIF51). Thirteen patients underwent OLIF51 (Group O) and 41 patients underwent MIS-TLIF (Group T) (Table 1). Since February 2020, amongst the OLIF51 surgeries performed, patient’s selection was based on absence of any paraspinal vessel anomalies and with a wider spine-vascular window of more than 20 mm and with no intervening peritoneal adhesions. Group O included 13 women (average 74.6 ± 3.2 years), while group T included 4 men and 38 women (average 70.5 ± 6.6 years). Mean follow-up period was 23.9 ± 7.0 months for group O and 28.9 ± 9.5 months for group T, ranging from 12 to 42 months.

### 2.1. Operation Procedure

#### 2.1.1. Primary Surgery (OLIF L1-S1 in Group O or OLIF L1-5 in Group T) (Figure 1) [9]

The patient was placed in the right lateral decubitus position on an adjustable hinged operating carbon fiber table (OSI Axis Jackson table; Mizuho, Union City, CA, USA) to perform CT scan by O-arm (Medtronic, Medtronic Sofamor Danek, Minneapolis, MN, USA). Axillary roll was placed to protect the neurovascular structures in the axilla. The patient should be in the center and the legs o should be slightly flexed to loosen the psoas muscles and the lumbar nerve plexus. The table was bent up to 15 degrees in convex to open the intervertebral disc space. The reference frame was attached percutaneously through the sacroiliac joint. For OLIF, neuromonitoring is not always necessary; however, we routinely use left flank approach with the patient in right lateral decubitus and neuromonitoring to prevent injury to the IVC and to assess occurrence of neurological complications respectively during OLIF, making this a preferred approach despite the presence of the left sided convexity. IVC injury. The 3D reconstructed images were obtained and transmitted to the Stealth station navigation system Spine 7^R^ (Medtronic, Medtronic Sofamor Danek, Minneapolis, MN, USA).

**Figure 1 medicina-59-00838-f001:**
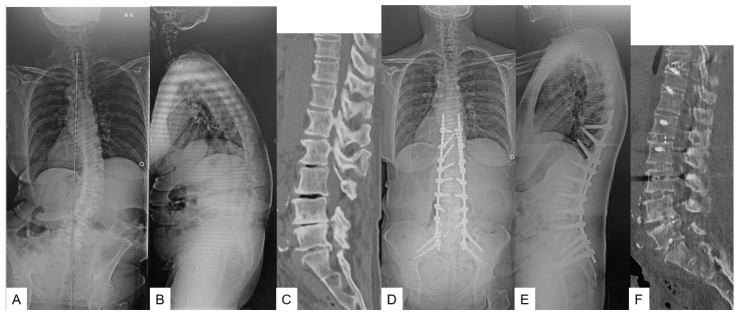
61-year-old female, adult spinal deformity, OLIF L1-S1 and T10-SAI. (**A**) Preoperative posteroanterior radiogram; (**B**) Preoperative lateral radiogram; Lateral radiogram showed severe sagittal malalignment; SVA of 181 mm, PT of 49°, PI-LL of 42°. (**C**) Preoperative CT. L5-S1 angle was 7.1°; (**D**) Postoperative posteroanterior radiogram; (**E**) Postoperative lateral radiogram. Lateral radiogram showed good sagittal malalignment; SVA of 5 mm, PT of 14°, PI-LL of 2°; (**F**) Postoperative CT. L5-S1 angle was 19.6°.

All the navigated spinal instrument were verified, following which, the best entry point for each of the disc spaces were marked by the navigated pin point probe. Ideally, 3 oblique skin incisions of approximately 5 cm were necessary for this technique. One was for both L1-L2 and L2-L3, another was for L3-L4 and L4-L5, and the other was L5-S1. For TLIF51 group, OLIF51 was not performed because of narrow vascular window or implant availability. The cages are to be placed from L1-2 to L5-S1 aided by the accuracy of navigation. For bone grafting, we used mixture of iliac bone and allograft. Usually, a single O arm scan was adequate to perform this procedure. After abdominal muscles were dissected and the disc space is exposed, at each level OLIF was performed with precisely calibrated navigated instruments.

#### 2.1.2. Secondary Surgery (T10-SAI in Group O or T10-SAI + TLIF51 in Group T) (Figure 1)

A week later, the patients were taken up for the secondary procedure where they were placed in a prone position on the Axis Jackson table. The reference frame was placed percutaneously around T11 spinous process and O-arm scan was done and the images were obtained from T10 to L3. After every navigated instrument was calibrated, PPS were inserted and place aided by navigation. For proximal T10 and T11, triangular fixation was ideal to prevent screw back out [masa]. After taking another O-arm scan from L4-S1, dual S2-sacral alar iliac (S2-SAI) screws were inserted to enhance the pelvic anchors [10].

For TLIF51 group, banana-shaped TLIF cage was inserted from left L5-S1 under navigation guidance. The anteroposterior and lateral radiograms should be obtained to check the correct placement of pedicle screws and S2-SAI screws. Following which, the Axis Jackson table was bent in concave by more than 20 degrees to create an adequate lumbar lordosis. The rods are usually pre-bent structurally to achieve an ideal lordotic contour and introduced percutaneously. The set screws were gradually tightened to create the required lumbar lordosis. In some cases, three or four rods technique was implemented to prevent rod breakage.

### 2.2. Clinical Assessment and Evaluation

In this study clinical presentation and symptomatic outcomes have been assessed using modalities including visual analogue scale (VAS) for back ache and Oswestry Disability Index (ODI). The clinical data was analyzed documented before the surgical procedure and at the end of 12, and 24 months following the surgery. All L1-5 level of both groups were performed by OLIF technique and only L5-S1 level was different in either of the groups. The duration of surgical procedure and the amount of blood loss were documented and for analysis in both groups and in the two stage operations.

The pre-operative and postsurgical complications such as, dural tears, end plate fracture, surgical site infection, epidural hematoma, focal sensory motor deficits reoperation, implant failure, malpositioned implants, nerve root injury and need for revision surgery were noted.

### 2.3. Radiographic Evaluation

Radiologically significant presentations were compared and analyzed preoperative and post operatively: sagittal vertical axis (SVA), pelvic incidence (PI)-lumbar lordosis (LL), pelvic tilt (PT), proximal junctional kyphosis (PJK), and screw back-out in a standing lateral spinal radiogram. L5-S1 disc height and L5-S1 angle, spinal bony union was evaluated in each group at the one-year follow-up using computer tomography (CT) (Figure 2).

### 2.4. Statistical Evaluation

The data that were analyzed, has been documented as mean ± standard deviation. For comparison between cohorts, Mann-Whitney U test analysis was utilized to calculate continuous variables, and chi-squared test was used to calculate dichotomous variables. McNemar’s test has been made use of for comparing the *p* values. A *p* value < 0.05 was defined as statistically prominent and notable.

## 3. Results

### 3.1. Patients Demographics

The demographic data is listed in Table 1. Patient’s age in numbers in group O was older than that in group T. Patient’s BMI of both groups were similar. Preoperative SVA, PT, and PI-LL were no difference in both groups. However, patient’s pelvic incidence (PI) in group O was larger than in group T. In group T, one patient had Parkinson disease and another patient had cerebral palsy as underlying medical condition.

### 3.2. Clinical Evaluation

Postoperative clinically significant data which has been documented are summarized in Table 2. Surgical time in group O was significantly less in group O than that in group T (356 ± 176 min vs. 492 ± 94 min, *p* = 0.003). Blood loss of both groups were almost equal (1016 ± 601 mL vs. 1252 ± 667 mL, *p* = 0.274) (Figure 3). Postoperative ODI in group O was a little better than that in group T but no statistically difference (22.4 ± 17.4% vs. 38.5 ± 20.4%, *p* = 0.131). VAS scores in both groups were similar (22.5 ± 6.5, 39.8 ± 7.4, *p* = 0.198). No vascular injury was observed in both groups. Two temporary neurological deterioration were observed in group T. Complication rates in both groups were no difference. Postoperative PJK were observed 38.9% and rod breakage were 9.3%. Surgical site infection was 3.7% and revision surgery was needed 18.5% of total cases.

### 3.3. Radiographic Evaluation

Radiographic parameters are analyzed and enumerated in Table 3. Postoperative L5-S1 angle gain in group O was statistically larger than that of group T (9.4 ± 4.7°, 1.6 ± 5.1°, *p* = 0.0001). Postoperative L5-S1 height gain in group O was better than that in group T (4.2 ± 2.9 mm, 0.8 ± 1.9 mm, *p* = 0.0002) (Figure 4). Postoperative SVA in both groups were improved postoperatively (43.7 ± 37.5 mm, 17.8 ± 37.8, *p* = 0.046). Postoperative PI-LL in both groups became normal values in both groups (8.69 ± 9.9°, 0.35 ± 12.8°, *p* = 0.025) in both groups. Postoperative average PT became normal in both group (17.7 ± 4.3°, 14.4 ± 9.8, *p* = 0.107). L5-S1 angle and height in group O were better than those in group T. The revision surgeries were performed for the rod breakage cases. Solid bony fusions were observed in all cases at final follow-up.

## 4. Discussion

Spinal malalignment in adulthood is called as adult spinal deformity [1,2]. These deformities occur due to degenerative or traumatic changes in intervertebral disc and facet joints [20]. These patients suffered medical, social, and psychological disability because of back pain and neurological symptoms [21]. Health impact studies using SF-36 score have found that score of pretreatment ASD patients is comparable to patients with diabetes, cancer, chronic heart disease and lung disease, limited vision [22,23]. Few decades ago, relatively limited literature was available on adult spine deformity as compared to adolescent scoliosis [24,25]. These patients were considered unfit for major spinal procedure due to lack of ideal surgical technique, poor bone quality, unavailability of proper implants [26]. Hence, these patients were counselled that they must live with this deformity and nothing much can be done. However, progress in anesthesia, surgical techniques and implants have made spine surgery in such patients possible [2,27].

Open surgery for ASD is effective in correcting deformity but is associated with complication rate as high 78% and major complications ranging from 10% to 55% [28,29]. Minimally invasive surgery on the other hand has comparable results with open surgery with less complication rates like dural tear, infections, less blood loss and need of transfusion, shorter hospital stay, early mobilization, less need of narcotic drugs [30,31,32]. Hence, use of MIS surgery is increasing for ASD correction.

It has been well demonstrated that 60% lumbar lordosis is located at L4-S1 segment [33,34]. So, reconstructing this while correcting ASD is important as studies have shown that patients with proximal junctional kyphosis showed more lordotic changes at upper lumbar level and smaller lordotic changes at lower lumbar level [35,36]. With OLIF51 greater correction of L5-S1 angle and height can be obtained as compared to TLIF51. This is due to wide opening of anterior disc space, anterior placement of cage and use of 12-degree lordotic cage in OLIF [37,38], Thus, creating more anatomic correction of lumbar lordosis.

Lumbar fusion can successfully be achieved using the PLIF technique. Nevertheless, these procedures are linked to problems such as damage to posterior support structures, paraspinal muscular injury, protracted muscle retraction, difficulties in intervertebral disc space visualization and endplate preparation and preparation, and the requirement for revision operations [39]. Because of these issues, surgeons have begun to employ an approach known as indirect decompression, which is dependent on the restoration of disc height. This has resulted in increases in foraminal height, as well as unnecessary strain and stretch of the ligamentum flavum and the posterior longitudinal ligament, both of which play an important role in stability of the central spinal canal [40]. Surgical techniques such as OLIF and lateral lumbar interbody fusion (LLIF) offer several benefits, including indirect decompression of nerve roots and other neural bodies, they can mandate a solid bony fusion, easier ways to insert and place a large sized cage, a significantly low risk of cage collapse, and a markedly low incidence of dural tears [40]. Other benefits include the ability to insert a larger cage.

The purpose of lumbar interbody fusion is to relieve segmental instability, decompress neuronal components, and preserve lumbar lordosis while causing as little harm as possible to the structures that are contiguous to the affected area. When using OLIF, indirect decompression can be achieved by positioning a cage with a large footprint; alternatively, if necessary, direct decompression can be carried out instead [41].

The use of a lateral retroperitoneal approach in OLIF51 results in several beneficial outcomes [42,43]. First, the abdominal structures shift downhill as a result of gravity while the patient is in the lateral position. As a direct consequence of this movement, the access corridor becomes larger. Because of this, it is feasible to approach the disc space with a lesser amount of peritoneal retraction compared to that which is required for ALIF. Second, it is possible to conduct long level lumbar interbody fusions from L1 to S1 in a single position because to the fact that both OLIF25 and OLIF51 employ the same lateral position without bending the hip. Third, a unilateral blunt dissection and retraction of the hypogastric sympathetic plexus in the OLIF51 approach is the same as that used to prevent hypogastric sympathetic plexus injury in the anterior approach to the L5-S1 level, and it can reduce the risk of postoperative retrograde ejaculation. This is the same technique that is used in the anterior approach to prevent hypogastric sympathetic plexus injury. In addition, the findings of Mun et al. [44] indicated that foraminal stenosis at the L5-S1 level was the most common kind of pathologic condition seen in both groups. OLIF51 appears to be particularly effective for the L5-S1 level, which is where foraminal stenosis is the primary issue. This is because foraminal stenosis is a favorable indication for indirect decompression through an increase in disc height.

Surgical time in our cases was 356 ± 176 min and 492 ± 94 min while blood loss was 1016 ± 601 mL and 1252 ± 667 mL for group O and group T respectively. Matsukura et al. [45] reported surgical time 535.9 ± 123.1 and 426.8 ± 96.2 and blood loss of 848.7 ± 477.1 and 2358.6 ± 1911.6 respectively for OLIF and TLIF group. It has been reported that mean surgical duration and blood loss are significantly lower in OLIF owing to use of muscle splitting approach and no need of laminectomy and facetectomy [18].

The results of Ohtori et al. [46] (250.35 min) were not comparable to the mean duration of OLIF, which was longer. Silvestre et al. [47] have theorized elements to minimize the length of OLIF by minimizing the use of a microscope and by utilizing solely bone replacement.

Postoperative VAS (mm) score was lesser among Group O (22.5 ± 6.5) compared to Group T (39.8 ± 7.4) though the difference was not statistically significant. Similar to our findings, Mun et al. [44] found that VAS, and ODI improved significantly in both OLIF51 and TLIF51 groups. There was no significant difference between the groups.

It has been proven beyond doubt that MIS surgery for ASD comparable clinical and radiological parameters [48]. In our study, preoperative and postoperative SVA, PI-LL, L5-S1 angle and L5-S1 height improved in both groups but group O demonstrated statistically significant improvement in these parameters as compared to group T (for values refer to table number 3). In a study by Park et al. [49] OLIF group showed improvement in SVA from 125.9 ± 21.3 to 27.1 ± 11.4, PI-LL from 36.5 ± 8.5 to 3.6 ± 3.0, L5-S1 angle from 9.8 ± 3.6 to 18.4 ± 3.7 while TLIF group showed improvement of SVA from 125.5 ± 22.1 to 32.7 ± 18.4, PI-LL from 34.1 ± 10.6 to 7.5 ± 3.2, L5-S1 angle from 7.5 ± 2.3 to 6.9 ± 2.8. Kotani et al. [50] demonstrated that OLIF51 showed disc height increase to 17 degrees while for TLIF51 it was 11.5 degrees. In study by Dorward et al. [51]. Preoperative SVA was 297 mm and 551 mm for TLIF group and OLIF group respectively which improved postoperatively to 172 mm (40.6% improvement) and 175 mm (68.9% improvement) for TLIF and OLIF group. Thus, our results are comparable with other studies.

Approach-related problems are another significant issue that has to be addressed with the OLIF technique [40,52]. The OLIF operation is performed via the retroperitoneal approach, which places the surgeon near to the abdominal arteries, psoas muscle, and ureter, among other anatomical structures. Spine surgeons are not familiar with these structures, which is why they are continually worried about the potential for approach-related problems. Chang et al. [13] found that the rate of complications due to the method of approach was comparable between the two groups. Compared to the OLIF approach, the TLIF technique had a greater risk of nerve damage, cage migration and subsidence, and hematoma in the prior research that investigated the many types of problems that might arise during the treatment. On the other hand, in comparison to the TLIF approach, the OLIF technique posed a greater threat of harm to the ureter, major arteries, sympathetic chain, and other retroperitoneal structures. Spine surgeons are worried about the risks associated with the OLIF operation, particularly the risk of harm to major arteries and other vital retroperitoneal structures [13].

However, the results of Chang et al. [13] indicated that the occurrence was exceedingly uncommon, and the rate of serious complications did not differ substantially between the two groups. Therefore, in terms of approach-related difficulties, the OLIF operation is identical to the TLIF method; consequently, undue anxiety over the OLIF technique is unneeded. TLIF is the procedure that was first developed. In the meanwhile, the surgical outcomes, such as blood loss, surgery time, and hospital stay, were taken into consideration and assessed in the meta-analysis. Chang et al. [13] reported that the OLIF group experienced much less blood loss and a shorter duration of hospital stay compared to the TLIF group; however, there was no significant difference in the amount of time spent performing the surgery between the two groups [13].

In our cases, we encountered some complications like PJK in 38.9% cases, rod breakage in 9.3%, surgical site infection in 3.7% and revision surgery in 18.5 cases. In another study incidence of PJK was 31.2%, implant failure was 10%, revision surgery was 7%, surgical site infection was 2.4% [45]. This shows that our findings are comparable with other studies.

Limitations of this research are, small cohort of patients, smaller number of OLIF group as compared to TLIF group which might give in skewed results after analysis. OLIF51 technique is relatively new one, so the period of both techniques had some difference. Retrospective method of study reduces power of study, prospective study with higher number of patients might be needed to have more clarity on this topic.

## 5. Conclusions

Clinical outcomes were not significantly different, but surgical time in OLIF51 was significantly less than that in TLIF51. The radiographic outcomes showed that OLIF51 created more L5-S1 lordosis and L5-S1 disc height compared with TLIF 51. C-arm free cMIS is a safe and effective technique that reduces radiation exposures. This latest procedure diminishes exposure of the surgeon and operation room staff to radiation as compared with traditional lumbar spine surgery techniques being practiced.

## Figures and Tables

**Figure 2 medicina-59-00838-f002:**
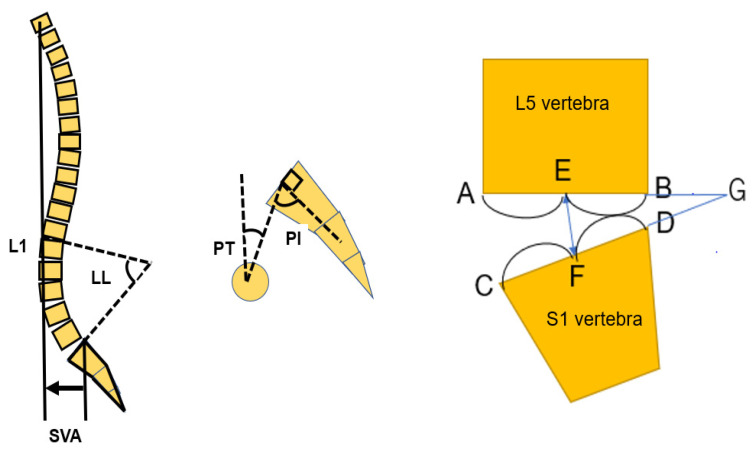
Spinopelvic parameters and L5-S1 parameters. Sagittal vertical axis (SVA), pelvic incidence (PI)-lumbar lordosis (LL), pelvic tilt (PT), L5-S1 angle; angle AGC, L5-S1 height; EF.

**Figure 3 medicina-59-00838-f003:**
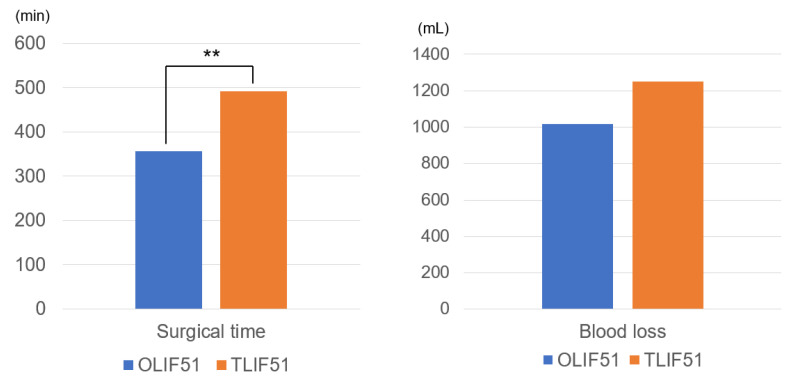
Surgical time and blood loss in both groups. ** *p* < 0.01.

**Figure 4 medicina-59-00838-f004:**
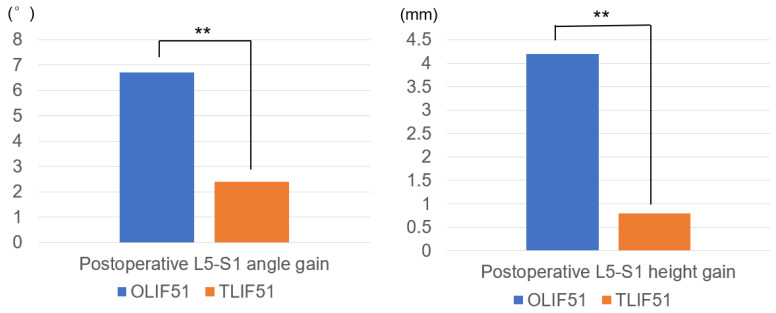
Postoperative L5-S1 angle and height gain in both groups at final follow-up. ** *p* < 0.01.

**Table 1 medicina-59-00838-t001:** Patient demographics.

	**Group O (*N* = 13)**	**Group T (*N* = 41)**	***p* Value**
L5-S1 fusion	OLIF51	MIS-TLIF51	
Patients	Man 0, Woman 13	Man 4, Woman 37	0.243
Age (year)	74.6 ± 3.2	70.5 ± 6.6	0.023 *
BMI (kg/m^2^)	22.7 ± 3.7	23.0 ± 4.2	0.715
SVA (mm)	100.8 ± 58.2	95.1 ± 55.8	0.934
PI (°)	57.5 ± 7.4	49.6 ± 11.3	0.035 *
PT (°)	39.2 ± 9.4	33.0 ± 11.2	0.062
PI-LL (°)	46.4 ± 21.3	37.0 ± 22.2	0.185
Disease	None	Parkinson 1, CP 1	

CP = Cerebral palsy. * *p* < 0.05.

**Table 2 medicina-59-00838-t002:** Clinical results of both groups at final follow-up.

	Group O (*N* = 13)	Group T (*N* = 41)	*p* Value
Surgical time (minutes)	356 ± 176	492 ± 94	0.003 **
Blood loss (mL)	1016 ± 601	1252 ± 667	0.274
Postoperative ODI (%)	22.4 ± 17.4	38.5 ± 20.4	0.131
Postoperative VAS (mm)	22.5 ± 6.5	39.8 ± 7.4	0.198
Complication			
PJK (+/−)	4/9	17/24	0.491
Rod breakage (+/−)	2/11	3/38	0.382
SSI (+/−)	1/12	1/40	0.400
Reoperation (+/−)	2/11	8/33	0.570

ODI: Oswestry disability index, VAS: Visual analog scale, PJK: Proximal junctional kyphosis, SSI: Surgical site infection. ** *p* < 0.01.

**Table 3 medicina-59-00838-t003:** Radiographic results of both groups at final follow-up.

	Group O (*N* = 13)	Group T (*N* = 41)	*p* Value
Preoperative L5-S1 angle (°)	12.0 ± 5.4	10.5 ± 4.7	0.459
Postoperative L5-S1 angle gain (°)	9.4 ± 4.7	1.6 ± 5.1	0.0001 **
Preoperative L5-S1 height (mm)	7.1 ± 2.2	8.6 ± 2.5	0.052
Postoperative L5-S1 height gain (mm)	4.2 ± 2.9	0.8 ± 1.9	0.0002 **
Preoperative SVA (mm)	100.8 ± 58.2	95.1 ± 55.8	0.943
Postoperative SVA (mm)	43.7 ± 37.5	17.8 ± 37.8	0.046 *
Preoperative PI-LL (°)	46.4 ± 21.3	37.0 ± 22.2	0.185
Postoperative PI-LL (°)	8.69 ± 9.9	0.35 ± 12.8	0.025 *
Preoperative PT (°)	39.2 ± 9.4	33.0 ± 11.2	0.062
Postoperative PT (°)	17.7 ± 4.3	14.4 ± 9.8	0.107

SVA: sagittal vertical axis, PI: pelvic incidence, LL: lumbar lordosis, PT: pelvic tilt. * *p* < 0.05, ** *p* < 0.01.

## Data Availability

The data presented in this study are available in the article.

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
