# Peer review of "Comparison of C-Arm-Free Oblique Lumbar Interbody Fusion L5-S1 (OLIF51) with Transforaminal Lumbar Interbody Fusion L5-S1 (TLIF51) for Adult Spinal Deformity"

_medicina, 2023, doi:10.3390/medicina59050838_

Round 1
Reviewer 1 Report
This is a article that compares "C-arm-free" OLIF and TLIF on L5-S1 disk in patients with Adult Spinal Deformity.
I have some concerns about this paper:
- the Autors retrospectivelly made a comparison between a group of patients treated for ASD using a navigated procedure (OLIF using O-arm) and a group of patients treated for ASD using TLIFs. In the text they didn't describe the TLIF-procedure , especially if they used O-arm or not.
- I presume that in the TLIF-patients the pedicle screws and the cages had been inserted in the same surgical procedure but the Authors didn't specify this in the text; so did the surgical times for OLIF consider also the time of the second surgery done after 1 week for putting the screws and rods (and eventually osteotomies), or not? This could better explain the difference in surgical times between OLIF and TLIF.
- the Authors didn't explain if the OLIF-group and the TLIF-group were operated during the same period of time (2016-2022) or if the OLIF-patients are the most recent patients they did; considering the difference in numbers bertween the groups this could be of importance in terms of learning curve or what else;
- in the text the Authors described the surgical procedure for OLIFs considering also L1-L2, L2-L3, L3-L4 but they didn't outline in a table these aspects in the evaluation of cases; especially for the TLIF group; obviously the more levels they did the more time they spent and the more blood loss they had. So how many 2, 3 4 or more levels did they treat?
Author Response
We appreciate your efforts and contribution.
To Reviewer 1
This is an article that compares "C-arm-free" OLIF and TLIF on L5-S1 disk in patients with Adult Spinal Deformity.
I have some concerns about this paper:
- the Autors retrospectivelly made a comparison between a group of patients treated for ASD using a navigated procedure (OLIF using O-arm) and a group of patients treated for ASD using TLIFs. In the text they didn't describe the TLIF-procedure , especially if they used O-arm or not.
We appreciate your important comments. We added the sentences as follows.
For TLIF51 group, banana-shaped TLIF cage was inserted from left L5-S1 under navigation guidance.
- I presume that in the TLIF-patients the pedicle screws and the cages had been inserted in the same surgical procedure but the Authors didn't specify this in the text; so did the surgical times for OLIF consider also the time of the second surgery done after 1 week for putting the screws and rods (and eventually osteotomies), or not? This could better explain the difference in surgical times between OLIF and TLIF.
Thank you for your thoughtful comments. The surgical time of both groups were anterior and posterior procedures together. All L1-5 level of both groups were performed by OLIF technique and only L5/S1 level was different in two groups.
We added the sentences as follows.
For TLIF51 group, OLIF51 was not performed because of narrow vascular window or implant availability. All L1-5 level of both groups were performed by OLIF technique and only L5/S1 level was different in two groups. The surgical time was evaluated by both two stage operations.
- the Authors didn't explain if the OLIF-group and the TLIF-group were operated during the same period of time (2016-2022) or if the OLIF-patients are the most recent patients they did; considering the difference in numbers bertween the groups this could be of importance in terms of learning curve or what else;
We appreciate your valuable comments. As you point out, OLIF51 technique is relatively new one, so OLIF51 group were mainly from 2019 to 2022. However, we had to choose MIS-TLIF because of the narrow vascular window for OLIF51. In limitation part, we added the sentence as follows.
OLIF51 technique is relatively new one, so the period of both techniques had some difference.
- in the text the Authors described the surgical procedure for OLIFs considering also L1-L2, L2-L3, L3-L4 but they didn't outline in a table these aspects in the evaluation of cases; especially for the TLIF group; obviously the more levels they did the more time they spent and the more blood loss they had. So how many 2, 3 4 or more levels did they treat?
Thank you for your important comments.
In this study, all patients underwent OLIF from L1 to L5.

Reviewer 2 Report
After review this study, my suggestion and comment are the following:
1. In Materials and Methods: Author should add more inclusion criteria for this study such as 1) Failure of conservative treatment, 2) severe spinal stenosis or others. Because we normally do not operate in patient with have only radiographic abnormality. In addition, in Table 1 author should give more details both clinical data and radiographic data such as underlying disease, BMI, number of levels of OLIF, SVA, Cobb angle, pelvic tilt in both groups.
2. In Materials and Methods: Author should calculate sample size base on previous study. Because there were many previous studies about lordotic angle comparison of OLIF5-1 and MIS TLIF.
3. In Materials and Methods: Author should give reason why author used MIS TLIF instead of OLIF in T group patient such as vascular deformity in which obstruct OLIF operation.
4. In Operation procedure: Author should give more details in type of bone substitutes (BMP-2 or HA??) in OLIF group and MIS TLIF group because it has effect on rod breakage and revision surgery.
5. In Operation procedure: Was author use right side approach for all patients even in patient with right side convex curve?
6. In figure 1: Was PEEK used in this patient OLIF PEEK? Why does author add screw fixation in this OLIF L5-S1 surgery. Normally we added screw in ALIF surgery for preventing PEEK migration.
7. In clinical evaluation: Author should give more details on early postoperative complications in both groups such as vascular injury, bowel ileus, or neurological complications.
8. In table 2: Author should describe timing of postoperative ODI and VAS (12 or 24 months postoperative?).
9. In table 3: Author should describe timing of radiographic finding (12 or 24 months postoperative?).
10. In radiographic evaluation: In one sentence “Solid bony fusions were observed in all cases”. Why did these both groups having rod breakage?
Author Response
We appreciate your efforts and contribution.
To Reviewer 2
After review this study, my suggestion and comment are the following:
- In Materials and Methods: Author should add more inclusion criteria for this study such as 1) Failure of conservative treatment, 2) severe spinal stenosis or others. Because we normally do not operate in patient with have only radiographic abnormality. In addition, in Table 1 author should give more details both clinical data and radiographic data such as underlying disease, BMI, number of levels of OLIF, SVA, Cobb angle, pelvic tilt in both groups.
We appreciate your important comments. According to your advice, we added the sentences as follows.
2) severe low back pain, difficulty of walking, and disturbance of active daily life, 3) failure of 2 months conservative treatment.
We changed table 1.
In Materials and Methods: Author should calculate sample size base on previous study. Because there were many previous studies about lordotic angle comparison of OLIF5-1 and MIS TLIF.
Thank you for your valuable comments. We’d like to increase the number of OLIF51 group. However, there was a statistically difference in the radiological results between two groups. So we added the following sentences in the limitation part.
The number of OLIF51 group was a little small.
- In Materials and Methods: Author should give reason why author used MIS TLIF instead of OLIF in T group patient such as vascular deformity in which obstruct OLIF operation.
We appreciate your thoughtful comments. As you point out, OLIF51 technique is relatively new one, so OLIF51 group were mainly from 2019 to 2022. However, we had to choose MIS-TLIF because of the narrow vascular window for OLIF51 from 2019 to 2022.
We added the sentences in materials and methods part as follows.
OLIF51 group were mainly from 2019 to 2022. However, we had to choose MIS-TLIF because of the narrow vascular window for OLIF51 from 2019 to 2022.
- In Operation procedure: Author should give more details in type of bone substitutes (BMP-2 or HA??) in OLIF group and MIS TLIF group because it has effect on rod breakage and revision surgery.
Thank you for your important comments. For bone grafts, we used mixture of iliac bone and allograft.
- In Operation procedure: Was author use right side approach for all patients even in patient with right side convex curve?
We appreciate your valuable comments. Yes. We always use left side approach for OLIF even it is convex side because of IVC.
- In figure 1: Was PEEK used in this patient OLIF PEEK? Why does author add screw fixation in this OLIF L5-S1 surgery. Normally we added screw in ALIF surgery for preventing PEEK migration.
Thank you for your comments. We used OLIF PEEK. For OLIF51 procedure, it is mandatory to put screws to prevent cage back out, which is the same reason of ALIF.
- In clinical evaluation: Author should give more details on early postoperative complications in both groups such as vascular injury, bowel ileus, or neurological complications.
We appreciate your thoughtful comments. We added the sentences in results part as follows.
No vascular injury was observed in both groups. Two temporary neurological deterioration were observed in group T.
In table 2: Author should describe timing of postoperative ODI and VAS (12 or 24 months postoperative?).
Thank you for your comment. These are mainly evaluated 24 months after surgery.
- In table 3: Author should describe timing of radiographic finding (12 or 24 months postoperative?).
Thank you for your comment. These are mainly evaluated 24 months after surgery.
- In radiographic evaluation: In one sentence “Solid bony fusions were observed in all cases”. Why did these both groups having rod breakage?
Thank you for your comments. We are sorry to make you misunderstand. Solid bony fusions were observed in all cases at the final follow-up. The revision surgeries were performed for the rod breakage cases.
We changed the sentence as follows.
The revision surgeries were performed for the rod breakage cases.
Solid bony fusions were observed in all cases at the final follow-up.

Round 2
Reviewer 2 Report
Dear, author
Thank you for your response to all my comments. I still have some more comments for improvment your manuscript.
1. In table 1 patient demographics: Normally, this table 1 should be included in Results section. Furthermore, author should add more sentences about BMI, PI, PT, PI-LL, and disease in your results section.
2. According to this sentence in your response to my previous comment
My comment: “4. In Operation procedure: Was author use right side approach for all patients even in patient with right side convex curve?”
Your response: “We appreciate your valuable comments. Yes. We always use left side approach for OLIF even it is convex side because of IVC.”
In your revised manuscript, you still use right lateral decubitus position. I confused which side that you always perform OLIF.
3. Could you add more detail about timing in titles of table 2 and table 3?
Author Response
Dear the excellent reviewer,
We appreciate your important comments.
1.In table 1 patient demographics: Normally, this table 1 should be included in Results section. Furthermore, author should add more sentences about BMI, PI, PT, PI-LL, and disease in your results section.
Thank you for your valuable comment. We moved those in the results section.
2.According to this sentence in your response to my previous comment
My comment: “4. In Operation procedure: Was author use right side approach for all patients even in patient with right side convex curve?”
Your response: “We appreciate your valuable comments. Yes. We always use left side approach for OLIF even it is convex side because of IVC.”
In your revised manuscript, you still use right lateral decubitus position. I confused which side that you always perform OLIF.
We are sorry to make you confused. We always use right lateral decubitus position, which means right side is on the table and left side is up.
We added the sentence as follows; even if the left side is convex side to avoid IVC injury.
3.Could you add more detail about timing in titles of table 2 and table 3?
We appreciate your important comment. We added the information according to your advice.